# Predictive Entropy Search for Efficient Global Optimization of Black-box Functions

**José Miguel Hernández-Lobato**
jmh233@cam.ac.uk
University of Cambridge

**Matthew W. Hoffman**
mwh30@cam.ac.uk
University of Cambridge

**Zoubin Ghahramani**
zoubin@eng.cam.ac.uk
University of Cambridge

## Abstract

We propose a novel information-theoretic approach for Bayesian optimization called Predictive Entropy Search (PES). At each iteration, PES selects the next evaluation point that maximizes the expected information gained with respect to the global maximum. PES codifies this intractable acquisition function in terms of the expected reduction in the differential entropy of the predictive distribution. This reformulation allows PES to obtain approximations that are both more accurate and efficient than other alternatives such as Entropy Search (ES). Furthermore, PES can easily perform a fully Bayesian treatment of the model hyperparameters while ES cannot. We evaluate PES in both synthetic and real-world applications, including optimization problems in machine learning, finance, biotechnology, and robotics. We show that the increased accuracy of PES leads to significant gains in optimization performance.

## 1 Introduction

Bayesian optimization techniques form a successful approach for optimizing black-box functions [5]. The goal of these methods is to find the global maximizer of a nonlinear and generally non-convex function $f$ whose derivatives are unavailable. Furthermore, the evaluations of $f$ are usually corrupted by noise and the process that queries $f$ can be computationally or economically very expensive. To address these challenges, Bayesian optimization devotes additional effort to modeling the unknown function $f$ and its behavior. These additional computations aim to minimize the number of evaluations that are needed to find the global optima.

Optimization problems are widespread in science and engineering and as a result so are Bayesian approaches to this problem. Bayesian optimization has successfully been used in robotics to adjust the parameters of a robot's controller to maximize gait speed and smoothness [16] as well as parameter tuning for computer graphics [6]. Another example application in drug discovery is to find the chemical derivative of a particular molecule that best treats a given disease [20]. Finally, Bayesian optimization can also be used to find optimal hyper-parameter values for statistical [29] and machine learning techniques [24].

As described above, we are interested in finding the global maximizer $\mathbf{x}_\star = \arg\max_{\mathbf{x} \in \mathcal{X}} f(\mathbf{x})$ of a function $f$ over some bounded domain, typically $\mathcal{X} \subset \mathbb{R}^d$. We assume that $f(\mathbf{x})$ can only be evaluated via queries to a black-box that provides noisy outputs of the form $y_i \sim \mathcal{N}(f(\mathbf{x}_i), \sigma^2)$. We note, however, that our framework can be extended to other non-Gaussian likelihoods. In this setting, we describe a sequential search algorithm that, after $n$ iterations, proposes to evaluate $f$ at some location $\mathbf{x}_{n+1}$. To make this decision the algorithm conditions on all previous observations $\mathcal{D}_n = \{(\mathbf{x}_1, y_1), \ldots, (\mathbf{x}_n, y_n)\}$. After $N$ iterations the algorithm makes a final recommendation $\widetilde{\mathbf{x}}_N$ for the global maximizer of the latent function $f$.

We take a Bayesian approach to the problem described above and use a probabilistic model for the latent function $f$ to guide the search and to select $\widetilde{\mathbf{x}}_N$. In this work we use a zero-mean Gaussian

| **Algorithm 1** Generic Bayesian optimization | **Algorithm 2** PES acquisition function |
|---|---|
| **Input:** a black-box with unknown mean $f$ | **Input:** a candidate $\mathbf{x}$; data $\mathcal{D}_n$ |
| 1: **for** $n = 1, \ldots, N$ **do** | 1: sample $M$ hyperparameter values $\{\boldsymbol{\psi}^{(i)}\}$ |
| 2:     select $\mathbf{x}_n = \arg\max_{\mathbf{x}\in\mathcal{X}} \alpha_{n-1}(\mathbf{x})$ | 2: **for** $i = 1, \ldots, M$ **do** |
| 3:     query the black-box at $\mathbf{x}_n$ to obtain $y_n$ | 3:     sample $f^{(i)} \sim p(f|\mathcal{D}_n, \boldsymbol{\phi}, \boldsymbol{\psi}^{(i)})$ |
| 4:     augment data $\mathcal{D}_n = \mathcal{D}_{n-1} \cup \{(\mathbf{x}_n, y_n)\}$ | 4:     set $\mathbf{x}_\star^{(i)} \leftarrow \arg\max_{\mathbf{x}\in\mathcal{X}} f^{(i)}(\mathbf{x})$ |
| 5: **end for** | 5:     compute $\mathbf{m}_0^{(i)}, \mathbf{V}_0^{(i)}$ and $\widetilde{\mathbf{m}}^{(i)}, \widetilde{\mathbf{v}}^{(i)}$ |
| 6: **return** $\widetilde{\mathbf{x}}_N = \arg\max_{\mathbf{x}\in\mathcal{X}} \mu_N(\mathbf{x})$ | 6:     compute $v_n^{(i)}(\mathbf{x})$ and $v_n^{(i)}(\mathbf{x}|\mathbf{x}_\star^{(i)})$ |
|  | 7: **end for** |
|  | 8: **return** $\alpha_n(\mathbf{x})$ as in (10) |

process (GP) prior for $f$ [22]. This prior is specified by a positive-definite kernel function $k(\mathbf{x}, \mathbf{x}')$. Given any finite collection of points $\{\mathbf{x}_1, \ldots, \mathbf{x}_n\}$, the values of $f$ at these points are jointly zero-mean Gaussian with covariance matrix $\mathbf{K}_n$, where $[\mathbf{K}_n]_{ij} = k(\mathbf{x}_i, \mathbf{x}_j)$. For the Gaussian likelihood described above, the vector of concatenated observations $\mathbf{y}_n$ is also jointly Gaussian with zero-mean. Therefore, at any location $\mathbf{x}$, the latent function $f(\mathbf{x})$ conditioned on past observations $\mathcal{D}_n$ is then Gaussian with marginal mean $\mu_n(\mathbf{x})$ and variance $v_n(\mathbf{x})$ given by

$$\mu_n(\mathbf{x}) = \mathbf{k}_n(\mathbf{x})^\mathsf{T}(\mathbf{K}_n + \sigma^2 \mathbf{I})^{-1}\mathbf{y}_n, \quad v_n(\mathbf{x}) = k(\mathbf{x}, \mathbf{x}) - \mathbf{k}_n(\mathbf{x})^\mathsf{T}(\mathbf{K}_n + \sigma^2 \mathbf{I})^{-1}\mathbf{k}_n(\mathbf{x}), \quad (1)$$

where $\mathbf{k}_n(\mathbf{x})$ is a vector of cross-covariance terms between $\mathbf{x}$ and $\{\mathbf{x}_1, \ldots, \mathbf{x}_n\}$.

Bayesian optimization techniques use the above predictive distribution $p(f(\mathbf{x})|\mathcal{D}_n)$ to guide the search for the global maximizer $\mathbf{x}_\star$. In particular, $p(f(\mathbf{x})|\mathcal{D}_n)$ is used during the computation of an acquisition function $\alpha_n(\mathbf{x})$ that is optimized at each iteration to determine the next evaluation location $\mathbf{x}_{n+1}$. This process is shown in Algorithm 1. Intuitively, the acquisition function $\alpha_n(\mathbf{x})$ should be high in areas where the maxima is most likely to lie given the current data. However, $\alpha_n(\mathbf{x})$ should also encourage exploration of the search space to guarantee that the recommendation $\widetilde{\mathbf{x}}_N$ is a global optimum of $f$, not just a global optimum of the posterior mean. Several acquisition functions have been proposed in the literature. Some examples are the probability of improvement (PI) [14], the expected improvement (EI) [19, 13] or upper confidence bounds (UCB) [26]. Alternatively, one can combine several of these acquisition functions [10].

The acquisition functions described above are based on probabilistic measures of improvement (PI an EI) or on optimistic estimates of the latent function (UCB) which implicitly trade off between exploiting the posterior mean and exploring based on the uncertainty. An alternate approach, introduced by [28], proposes maximizing the expected posterior information gain about the global maximizer $\mathbf{x}_\star$ evaluated over a grid in the input space. A similar strategy was later employed by [9] which although it requires no such grid, instead relies on a difficult-to-evaluate approximation. In Section 2 we derive a rearrangement of this information-based acquisition function which leads to a more straightforward approximation that we call Predictive Entropy Search (PES). In Section 3 we show empirically that our approximation is more accurate than that of [9]. We evaluate this claim on both synthetic and real-world problems and further show that this leads to real gains in performance.

## 2 Predictive entropy search

We propose to follow the information-theoretic method for active data collection described in [17]. We are interested in maximizing information about the location $\mathbf{x}_\star$ of the global maximum, whose posterior distribution is $p(\mathbf{x}_\star|\mathcal{D}_n)$. Our current information about $\mathbf{x}_\star$ can be measured in terms of the negative differential entropy of $p(\mathbf{x}_\star|\mathcal{D}_n)$. Therefore, our strategy is to select $\mathbf{x}_{n+1}$ which maximizes the expected reduction in this quantity. The corresponding acquisition function is

$$\alpha_n(\mathbf{x}) = \mathrm{H}[p(\mathbf{x}_\star|\mathcal{D}_n)] - \mathbb{E}_{p(y|\mathcal{D}_n, \mathbf{x})}[\mathrm{H}[p(\mathbf{x}_\star|\mathcal{D}_n \cup \{(\mathbf{x}, y)\})]], \quad (2)$$

where $\mathrm{H}[p(\mathbf{x})] = -\int p(\mathbf{x}) \log p(\mathbf{x}) d\mathbf{x}$ represents the differential entropy of its argument and the expectation above is taken with respect to the posterior predictive distribution of $y$ given $\mathbf{x}$. The exact evaluation of (2) is infeasible in practice. The main difficulties are i) $p(\mathbf{x}_\star|\mathcal{D}_n \cup \{(\mathbf{x}, y)\})$ must be computed for many different values of $\mathbf{x}$ and $y$ during the optimization of (2) and ii) the entropy computations themselves are not analytical. In practice, a direct evaluation of (2) is only

possible after performing many approximations [9]. To avoid this, we follow the approach described in [11] by noting that (2) can be equivalently written as the mutual information between $\mathbf{x}_\star$ and $y$ given $\mathcal{D}_n$. Since the mutual information is a symmetric function, $\alpha_n(\mathbf{x})$ can be rewritten as

$$\alpha_n(\mathbf{x}) = \mathrm{H}[p(y|\mathcal{D}_n, \mathbf{x})] - \mathbb{E}_{p(\mathbf{x}_\star|\mathcal{D}_n)}[\mathrm{H}[p(y|\mathcal{D}_n, \mathbf{x}, \mathbf{x}_\star)]], \tag{3}$$

where $p(y|\mathcal{D}_n, \mathbf{x}, \mathbf{x}_\star)$ is the posterior predictive distribution for $y$ given the observed data $\mathcal{D}_n$ and the location of the global maximizer of $f$. Intuitively, conditioning on the location $\mathbf{x}_\star$ pushes the posterior predictions up in locations around $\mathbf{x}_\star$ and down in regions away from $\mathbf{x}_\star$. Note that, unlike the previous formulation, this objective is based on the entropies of predictive distributions, which are analytic or can be easily approximated, rather than on the entropies of distributions on $\mathbf{x}_\star$ whose approximation is more challenging.

The first term in (3) can be computed analytically using the posterior marginals for $f(\mathbf{x})$ in (1), that is, $\mathrm{H}[p(y|\mathcal{D}_n, \mathbf{x})] = 0.5 \log[2\pi e\, (v_n(\mathbf{x}) + \sigma^2)]$, where we add $\sigma^2$ to $v_n(\mathbf{x})$ because $y$ is obtained by adding Gaussian noise with variance $\sigma^2$ to $f(\mathbf{x})$. The second term, on the other hand, must be approximated. We first approximate the expectation in (3) by averaging over samples $\mathbf{x}_\star^{(i)}$ drawn approximately from $p(\mathbf{x}_\star|\mathcal{D}_n)$. For each of these samples, we then approximate the corresponding entropy function $\mathrm{H}[p(y|\mathcal{D}_n, \mathbf{x}, \mathbf{x}_\star^{(i)})]$ using expectation propagation [18]. The code for all these operations is publicly available at `http://jmhl.org`.

## 2.1 Sampling from the posterior over global maxima

In this section we show how to approximately sample from the conditional distribution of the global maximizer $\mathbf{x}_\star$ given the observed data $\mathcal{D}_n$, that is,

$$p(\mathbf{x}_\star|\mathcal{D}_n) = p\big(f(\mathbf{x}_\star) = \max_{\mathbf{x} \in \mathcal{X}} f(\mathbf{x})\big|\mathcal{D}_n\big). \tag{4}$$

If the domain $\mathcal{X}$ is restricted to some finite set of $m$ points, the latent function $f$ takes the form of an $m$-dimensional vector $\mathbf{f}$. The probability that the $i$th element of $\mathbf{f}$ is optimal can then be written as $\int p(\mathbf{f}|\mathcal{D}_n) \prod_{j \le m} \mathbb{I}[f_i \ge f_j]\, d\mathbf{f}$. This suggests the following generative process: i) draw a sample from the posterior distribution $p(\mathbf{f}|\mathcal{D}_n)$ and ii) return the index of the maximum element in the sampled vector. This process is known as Thompson sampling or probability matching when used as an arm-selection strategy in multi-armed bandits [8]. This same approach could be used for sampling the maximizer over a continuous domain $\mathcal{X}$. At first glance this would require constructing an infinite-dimensional object representing the function $f$. To avoid this, one could sequentially construct $f$ while it is being optimized. However, evaluating such an $f$ would ultimately have cost $\mathcal{O}(m^3)$ where $m$ is the number of function evaluations necessary to find the optimum. Instead, we propose to sample and optimize an analytic approximation to $f$. We will briefly derive this approximation below, but more detail is given in Appendix A.

Given a shift-invariant kernel $k$, Bochner's theorem [4] asserts the existence of its Fourier dual $s(\mathbf{w})$, which is equal to the spectral density of $k$. Letting $p(\mathbf{w}) = s(\mathbf{w})/\alpha$ be the associated normalized density, we can write the kernel as the expectation

$$k(\mathbf{x}, \mathbf{x}') = \alpha\, \mathbb{E}_{p(\mathbf{w})}[e^{-i\mathbf{w}^\mathsf{T}(\mathbf{x} - \mathbf{x}')}] = 2\alpha\, \mathbb{E}_{p(\mathbf{w}, b)}[\cos(\mathbf{w}^\mathsf{T}\mathbf{x} + b)\cos(\mathbf{w}^\mathsf{T}\mathbf{x}' + b)], \tag{5}$$

where $b \sim \mathcal{U}[0, 2\pi]$. Let $\phi(\mathbf{x}) = \sqrt{2\alpha/m}\cos(\mathbf{W}\mathbf{x} + \mathbf{b})$ denote an $m$-dimensional feature mapping where $\mathbf{W}$ and $\mathbf{b}$ consist of $m$ stacked samples from $p(\mathbf{w}, b)$. The kernel $k$ can then be approximated by the inner product of these features, $k(\mathbf{x}, \mathbf{x}') \approx \phi(\mathbf{x})^\mathsf{T}\phi(\mathbf{x}')$. This approach was used by [21] as an approximation method in the context of kernel methods. The feature mapping $\phi(\mathbf{x})$ allows us to approximate the Gaussian process prior for $f$ with a linear model $f(\mathbf{x}) = \phi(\mathbf{x})^\mathsf{T}\boldsymbol{\theta}$ where $\boldsymbol{\theta} \sim \mathcal{N}(\mathbf{0}, \mathbf{I})$ is a standard Gaussian. By conditioning on $\mathcal{D}_n$, the posterior for $\boldsymbol{\theta}$ is also multivariate Gaussian, $\boldsymbol{\theta}|\mathcal{D}_n \sim \mathcal{N}(\mathbf{A}^{-1}\boldsymbol{\Phi}^\mathsf{T}\mathbf{y}_n, \sigma^2\mathbf{A}^{-1})$ where $\mathbf{A} = \boldsymbol{\Phi}^\mathsf{T}\boldsymbol{\Phi} + \sigma^2\mathbf{I}$ and $\boldsymbol{\Phi}^\mathsf{T} = [\phi(\mathbf{x}_1)\ldots\phi(\mathbf{x}_n)]$.

Let $\phi^{(i)}$ and $\boldsymbol{\theta}^{(i)}$ be a random set of features and the corresponding posterior weights sampled both according to the generative process given above. They can then be used to construct the function $f^{(i)}(\mathbf{x}) = \phi^{(i)}(\mathbf{x})^\mathsf{T}\boldsymbol{\theta}^{(i)}$, which is an approximate posterior sample of $f$—albeit one with a finite parameterization. We can then maximize this function to obtain $\mathbf{x}_\star^{(i)} = \arg\max_{\mathbf{x} \in \mathcal{X}} f^{(i)}(\mathbf{x})$, which is approximately distributed according to $p(\mathbf{x}_\star|\mathcal{D}_n)$. Note that for early iterations when $n < m$, we can efficiently sample $\boldsymbol{\theta}^{(i)}$ with cost $\mathcal{O}(n^2 m)$ using the method described in Appendix B.2 of [23]. This allows us to use a large number of features in $\phi^{(i)}(\mathbf{x})$.

## 2.2 Approximating the predictive entropy

We now show how to approximate $\mathrm{H}[p(y|\mathcal{D}_n, \mathbf{x}, \mathbf{x}_\star)]$ in (3). Note that we can write the argument to H in this expression as $p(y|\mathcal{D}_n, \mathbf{x}, \mathbf{x}_\star) = \int p(y|f(\mathbf{x}))p(f(\mathbf{x})|\mathcal{D}_n, \mathbf{x}_\star) \, df(\mathbf{x})$. Here $p(f(\mathbf{x})|\mathcal{D}_n, \mathbf{x}_\star)$ is the posterior distribution on $f(\mathbf{x})$ given $\mathcal{D}_n$ and the location $\mathbf{x}_\star$ of the global maximizer of $f$. When the likelihood $p(y|f(\mathbf{x}))$ is Gaussian, we have that $p(f(\mathbf{x})|\mathcal{D}_n)$ is analytically tractable since it is the predictive distribution of a Gaussian process. However, by further conditioning on the location $\mathbf{x}_\star$ of the global maximizer we are introducing additional constraints, namely that $f(\mathbf{z}) \leq f(\mathbf{x}_\star)$ for all $\mathbf{z} \in \mathcal{X}$. These constraints make $p(f(\mathbf{x})|\mathcal{D}_n, \mathbf{x}_\star)$ intractable. To circumvent this difficulty, we instead use the following simplified constraints:

C1. $\mathbf{x}_\star$ **is a local maximum.** This is achieved by letting $\nabla f(\mathbf{x}_\star) = \mathbf{0}$ and ensuring that $\nabla^2 f(\mathbf{x}_\star)$ is negative definite. We further assume that the non-diagonal elements of $\nabla^2 f(\mathbf{x}_\star)$, denoted by $\mathrm{upper}[\nabla^2 f(\mathbf{x}_\star)]$, are known, for example they could all be zero. This simplifies the negative-definite constraint. We denote by C1.1 the constraint given by $\nabla f(\mathbf{x}_\star) = \mathbf{0}$ and $\mathrm{upper}[\nabla^2 f(\mathbf{x}_\star)] = \mathbf{0}$. We denote by C1.2 the constraint that forces the elements of $\mathrm{diag}[\nabla^2 f(\mathbf{x}_\star)]$ to be negative.

C2. $f(\mathbf{x}_\star)$ **is larger than past observations.** We also assume that $f(\mathbf{x}_\star) \geq f(\mathbf{x}_i)$ for all $i \leq n$. However, we only observe $f(\mathbf{x}_i)$ noisily via $y_i$. To avoid making inference on these latent function values, we approximate the above hard constraints with the soft constraint $f(\mathbf{x}_\star) > y_{\max} + \epsilon$, where $\epsilon \sim \mathcal{N}(0, \sigma^2)$ and $y_{\max}$ is the largest $y_i$ seen so far.

C3. $f(\mathbf{x})$ **is smaller than** $f(\mathbf{x}_\star)$. This simplified constraint only conditions on the given $\mathbf{x}$ rather than requiring $f(\mathbf{x}_\star) \leq f(\mathbf{z})$ for all $\mathbf{z} \in \mathcal{X}$.

We incorporate these simplified constraints into $p(f(\mathbf{x})|\mathcal{D}_n)$ to approximate $p(f(\mathbf{x})|\mathcal{D}_n, \mathbf{x}_\star)$. This is achieved by multiplying $p(f(\mathbf{x})|\mathcal{D}_n)$ with specific factors that encode the above constraints. In what follows we briefly show how to construct these factors; more detail is given in Appendix B.

Consider the latent variable $\mathbf{z} = [f(\mathbf{x}_\star); \mathrm{diag}[\nabla^2 f(\mathbf{x}_\star)]]$. To incorporate constraint C1.1 we can condition on the data and on the "observations" given by the constraints $\nabla f(\mathbf{x}_\star) = \mathbf{0}$ and $\mathrm{upper}[\nabla^2 f(\mathbf{x}_\star)] = \mathbf{0}$. Since $f$ is distributed according to a GP, the joint distribution between $\mathbf{z}$ and these observations is multivariate Gaussian. The covariance between the noisy observations $\mathbf{y}_n$ and the extra noise-free derivative observations can be easily computed [25]. The resulting conditional distribution is also multivariate Gaussian with mean $\mathbf{m}_0$ and covariance $\mathbf{V}_0$. These computations are similar to those performed in (1). Constraints C1.2 and C2 can then be incorporated by writing

$$p(\mathbf{z}|\mathcal{D}_n, \mathrm{C1}, \mathrm{C2}) \propto \Phi_{\sigma^2}(f(\mathbf{x}_\star) - y_{\max}) \Big[ \textstyle\prod_{i=1}^{d} \mathbb{I}\big([\nabla^2 f(\mathbf{x}_\star)]_{ii} \leq 0\big) \Big] \mathcal{N}(\mathbf{z}|\mathbf{m}_0, \mathbf{V}_0), \qquad (6)$$

where $\Phi_{\sigma^2}$ is the cdf of a zero-mean Gaussian distribution with variance $\sigma^2$. The first new factor in this expression guarantees that $f(\mathbf{x}_\star) > y_{\max} + \epsilon$, where we have marginalized $\epsilon$ out, and the second set of factors guarantees that the entries in $\mathrm{diag}[\nabla^2 f(\mathbf{x}_\star)]$ are negative.

Later integrals that make use of $p(\mathbf{z}|\mathcal{D}_n, \mathrm{C1}, \mathrm{C2})$, however, will not admit a closed-form expression. As a result we compute a Gaussian approximation $q(\mathbf{z})$ to this distribution using Expectation Propagation (EP) [18]. The resulting algorithm is similar to the implementation of EP for binary classification with Gaussian processes [22]. EP approximates each non-Gaussian factor in (6) with a Gaussian factor whose mean and variance are $\widetilde{m}_i$ and $\widetilde{v}_i$, respectively. The EP approximation can then be written as $q(\mathbf{z}) \propto [\prod_{i=1}^{d+1} \mathcal{N}(z_i|\widetilde{m}_i, \widetilde{v}_i)]\mathcal{N}(\mathbf{z}|\mathbf{m}_0, \mathbf{V}_0)$. Note that these computations have so far not depended on $\mathbf{x}$, so we can compute $\{\mathbf{m}_0, \mathbf{V}_0, \widetilde{\mathbf{m}}, \widetilde{\mathbf{v}}\}$ once and store them for later use, where $\widetilde{\mathbf{m}} = (\widetilde{m}_1, \ldots, \widetilde{m}_{d+1})$ and $\widetilde{\mathbf{v}} = (\widetilde{v}_1, \ldots, \widetilde{v}_{d+1})$.

We will now describe how to compute the predictive variance of some latent function value $f(\mathbf{x})$ given these constraints. Let $\mathbf{f} = [f(\mathbf{x}); f(\mathbf{x}_\star)]$ be a vector given by the concatenation of the values of the latent function at $\mathbf{x}$ and $\mathbf{x}_\star$. The joint distribution between $\mathbf{f}$, $\mathbf{z}$, the evaluations $\mathbf{y}_n$ collected so far and the derivative "observations" $\nabla f(\mathbf{x}_\star) = \mathbf{0}$ and $\mathrm{upper}[\nabla^2 f(\mathbf{x}_\star)] = \mathbf{0}$ is multivariate Gaussian. Using $q(\mathbf{z})$, we then obtain the following approximation:

$$p(\mathbf{f}|\mathcal{D}_n, \mathrm{C1}, \mathrm{C2}) \approx \int p(\mathbf{f}|\mathbf{z}, \mathcal{D}_n, \mathrm{C1.1}) \, q(\mathbf{z}) \, d\mathbf{z} = \mathcal{N}(\mathbf{f}|\mathbf{m}_{\mathbf{f}}, \mathbf{V}_{\mathbf{f}}). \qquad (7)$$

Implicitly we are assuming above that $\mathbf{f}$ depends on our observations and constraint C1.1, but is independent of C1.2 and C2 given $\mathbf{z}$. The computations necessary to obtain $\mathbf{m}_{\mathbf{f}}$ and $\mathbf{V}_{\mathbf{f}}$ are similar

to those used above and in (1). The required quantities are similar to the ones used by EP to make predictions in the Gaussian process binary classifier [22]. We can then incorporate C3 by multiplying $\mathcal{N}(\mathbf{f}|\mathbf{m_f}, \mathbf{V_f})$ with a factor that guarantees $f(\mathbf{x}) < f(\mathbf{x}_\star)$. The predictive distribution for $f(\mathbf{x})$ given $\mathcal{D}_n$ and all the constraints can be approximated as

$$p(f(\mathbf{x})|\mathcal{D}_n, C1, C2, C3) \approx Z^{-1} \int \mathbb{I}(f_1 < f_2) \mathcal{N}(\mathbf{f}|\mathbf{m_f}, \mathbf{V_f}) \, df_2 \,, \tag{8}$$

where $Z$ is a normalization constant. The variance of the right hand size of (8) is given by

$$v_n(\mathbf{x}|\mathbf{x}_\star) = [\mathbf{V_f}]_{1,1} - v^{-1}\beta(\beta+\alpha)\{[\mathbf{V_f}]_{1,1} - [\mathbf{V_f}]_{1,2}\}^2 \,, \tag{9}$$

where $v = [-1, 1]^\mathsf{T}\mathbf{V_f}[-1, 1]$, $\alpha = m/\sqrt{v}$, $m = [-1, 1]^\mathsf{T}\mathbf{m_f}$, $\beta = \phi(\alpha)/\Phi(\alpha)$, and $\phi(\cdot)$ and $\Phi(\cdot)$ are the standard Gaussian density function and cdf, respectively. By further approximating (8) by a Gaussian distribution with the same mean and variance we can write the entropy as $\mathrm{H}[p(y|\mathcal{D}_n, \mathbf{x}, \mathbf{x}_\star)] \approx 0.5 \log[2\pi e(v_n(\mathbf{x}|\mathbf{x}_\star) + \sigma^2)]$.

The computation of (9) can be numerically unstable when $s$ is very close to zero. This occurs when $[\mathbf{V_f}]_{1,1}$ is very similar to $[\mathbf{V_f}]_{1,2}$. To avoid these numerical problems, we multiply $[\mathbf{V_f}]_{1,2}$ by the largest $0 \le \kappa \le 1$ that guarantees that $s > 10^{-10}$. This can be understood as slightly reducing the amount of dependence between $f(\mathbf{x})$ and $f(\mathbf{x}_\star)$ when $\mathbf{x}$ is very close to $\mathbf{x}_\star$. Finally, fixing $\mathrm{upper}[\nabla^2 f(\mathbf{x}_\star)]$ to be zero can also produce poor predictions when the actual $f$ does not satisfy this constraint. To avoid this, we instead fix this quantity to $\mathrm{upper}[\nabla^2 f^{(i)}(\mathbf{x}_\star)]$, where $f^{(i)}$ is the $i$th sample function optimized in Section 2.1 to sample $\mathbf{x}_\star^{(i)}$.

## 2.3 Hyperparameter learning and the PES acquisition function

We now show how the previous approximations are integrated to compute the acquisition function used by predictive entropy search (PES). This acquisition function performs a formal treatment of the hyperparameters. Let $\boldsymbol{\psi}$ denote a vector of hyperparameters which includes any kernel parameters as well as the noise variance $\sigma^2$. Let $p(\boldsymbol{\psi}|\mathcal{D}_n) \propto p(\boldsymbol{\psi}) p(\mathcal{D}_n|\boldsymbol{\psi})$ denote the posterior distribution over these parameters where $p(\boldsymbol{\psi})$ is a hyperprior and $p(\mathcal{D}_n|\boldsymbol{\psi})$ is the GP marginal likelihood. For a fully Bayesian treatment of $\boldsymbol{\psi}$ we must marginalize the acquisition function (3) with respect to this posterior. The corresponding integral has no analytic expression and must be approximated using Monte Carlo. This approach is also taken in [24].

We draw $M$ samples $\{\boldsymbol{\psi}^{(i)}\}$ from $p(\boldsymbol{\psi}|\mathcal{D}_n)$ using slice sampling [27]. Let $\mathbf{x}_\star^{(i)}$ denote a sampled global maximizer drawn from $p(\mathbf{x}_\star|\mathcal{D}_n, \boldsymbol{\psi}^{(i)})$ as described in Section 2.1. Furthermore, let $v_n^{(i)}(\mathbf{x})$ and $v_n^{(i)}(\mathbf{x}|\mathbf{x}_\star^{(i)})$ denote the predictive variances computed as described in Section 2.2 when the model hyperparameters are fixed to $\boldsymbol{\psi}^{(i)}$. We then write the marginalized acquisition function as

$$\alpha_n(\mathbf{x}) = \frac{1}{M} \sum_{i=1}^M \left\{ 0.5 \log[v_n^{(i)}(\mathbf{x}) + \sigma^2] - 0.5 \log[v_n^{(i)}(\mathbf{x}|\mathbf{x}_\star^{(i)}) + \sigma^2] \right\} \,. \tag{10}$$

Note that PES is effectively marginalizing the original acquisition function (2) over $p(\boldsymbol{\psi}|\mathcal{D}_n)$. This is a significant advantage with respect to other methods that optimize the same information-theoretic acquisition function but do not marginalize over the hyper-parameters. For example, the approach of [9] approximates (2) only for fixed $\boldsymbol{\psi}$. The resulting approximation is computationally very expensive and recomputing it to average over multiple samples from $p(\boldsymbol{\psi}|\mathcal{D}_n)$ is infeasible in practice.

Algorithm 2 shows pseudo-code for computing the PES acquisition function. Note that most of the computations necessary for evaluating (10) can be done independently of the input $\mathbf{x}$, as noted in the pseudo-code. This initial cost is dominated by a matrix inversion necessary to pre-compute $\mathbf{V}$ for each hyperparameter sample. The resulting complexity is $\mathcal{O}[M(n+d+d(d-1)/2)^3]$. This cost can be reduced to $\mathcal{O}[M(n+d)^3]$ by ignoring the derivative observations imposed on $\mathrm{upper}[\nabla^2 f(\mathbf{x}_\star)]$ by constraint C1.1. Nevertheless, in the problems that we consider $d$ is very small (less than 20). After these precomputations are done, the evaluation of (10) is $\mathcal{O}[M(n+d+d(d-1)/2)]$.

## 3 Experiments

In our experiments, we use Gaussian process priors for $f$ with squared-exponential kernels $k(\mathbf{x}, \mathbf{x}') = \gamma^2 \exp\{-0.5 \sum_i (x_i - x_i')^2/\ell_i^2\}$. The corresponding spectral density is zero-mean Gaussian with covariance given by $\mathrm{diag}([\ell_i^{-2}])$ and normalizing constant $\alpha = \gamma^2$. The model hyperparameters are $\{\gamma, \ell_1, \ldots, \ell_d, \sigma^2\}$. We use broad, uninformative Gamma hyperpriors.

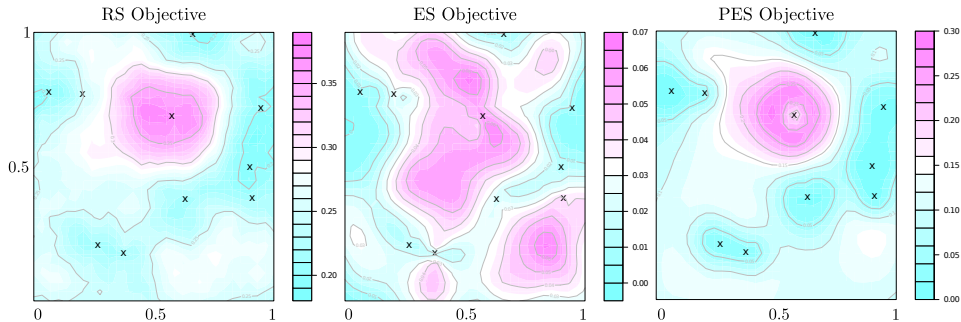

Figure 1: Comparison of different estimates of the objective function $\alpha_n(\mathbf{x})$ given by (2). Left, ground truth obtained by the rejection sampling method RS. Middle, approximation produced by the ES method. Right, approximation produced by the proposed PES method. These plots show that the PES objective is much more similar to the RS ground truth than the ES objective.

First, we analyze the accuracy of PES in the task of approximating the differential entropy (2). We compare the PES approximation (10), with the approximation used by the entropy search (ES) method [9]. We also compare with the ground truth for (2) obtained using a rejection sampling (RS) algorithm based on (3). For this experiment we generate the data $\mathcal{D}_n$ using an objective function $f$ sampled from the Gaussian process prior as in [9]. The domain $\mathcal{X}$ of $f$ is fixed to be $[0, 1]^2$ and data are generated using $\gamma^2 = 1$, $\sigma^2 = 10^{-6}$, and $\ell_i^2 = 0.1$. To compute (10) we avoid sampling the hyperparameters and use the known values directly. We further fix $M = 200$ and $m = 1000$.

The ground truth rejection sampling scheme works as follows. First, $\mathcal{X}$ is discretized using a uniform grid. The expectation with respect to $p(\mathbf{x}_\star|\mathcal{D}_n)$ in (3) is then approximated using sampling. For this, we sample $\mathbf{x}_\star$ by evaluating a random sample from $p(f|\mathcal{D}_n)$ on each grid cell and then selecting the cell with highest value. Given $\mathbf{x}_\star$, we then approximate $\mathrm{H}[p(y|\mathcal{D}_n, \mathbf{x}, \mathbf{x}_\star)]$ by rejection sampling. We draw samples from $p(f|\mathcal{D}_n)$ and reject those whose corresponding grid cell with highest value is not $\mathbf{x}_\star$. Finally, we approximate $\mathrm{H}[p(y|\mathcal{D}_n, \mathbf{x}, \mathbf{x}_\star)]$ by first, adding zero-mean Gaussian noise with variance $\sigma^2$ to the the evaluations at $\mathbf{x}$ of the functions not rejected during the previous step and second, we estimate the differential entropy of the resulting samples using kernels [1].

Figure 1 shows the objective functions produced by RS, ES and PES for a particular $\mathcal{D}_n$ with 10 measurements whose locations are selected uniformly at random in $[0, 1]^2$. The locations of the collected measurements are displayed with an "x" in the plots. The particular objective function used to generate the measurements in $\mathcal{D}_n$ is displayed in the left part of Figure 2. The plots in Figure 1 show that the PES approximation to (2) is more similar to the ground truth given by RS than the approximation produced by ES. In this figure we also see a discrepancy between RS and PES at locations near $\mathbf{x} = (0.572, 0.687)$. This difference is an artifact of the discretization used in RS. By zooming in and drawing many more samples we would see the same behavior in both plots.

We now evaluate the performance of PES in the task of finding the optimum of synthetic black-box objective functions. For this, we reproduce the within-model comparison experiment described in [9]. In this experiment we optimize objective functions defined in the 2-dimensional unit domain $\mathcal{X} = [0, 1]^2$. Each objective function is generated by first sampling 1024 function values from the GP prior assumed by PES, using the same $\gamma^2$, $\ell_i$ and $\sigma^2$ as in the previous experiment. The objective function is then given by the resulting GP posterior mean. We generated a total of 1000 objective functions by following this procedure. The left plot in Figure 2 shows an example function.

In these experiments we compared the performance of PES with that of ES [9] and expected improvement (EI) [13], a widely used acquisition function in the Bayesian optimization literature. We again assume that the optimal hyper-parameter values are known to all methods. Predictive performance is then measured in terms of the immediate regret (IR) $|f(\widetilde{\mathbf{x}}_n) - f(\mathbf{x}_\star)|$, where $\mathbf{x}_\star$ is the known location of the global maximum and $\widetilde{\mathbf{x}}_n$ is the recommendation of each algorithm had we stopped at step $n$—for all methods this is given by the maximizer of the posterior mean. The right plot in Figure 2 shows the decimal logarithm of the median of the IR obtained by each method across the 1000 *different* objective functions. Confidence bands equal to one standard deviation are obtained using the bootstrap method. Note that while averaging these results is also interesting, corresponding to the expected performance averaged over the prior, here we report the median IR

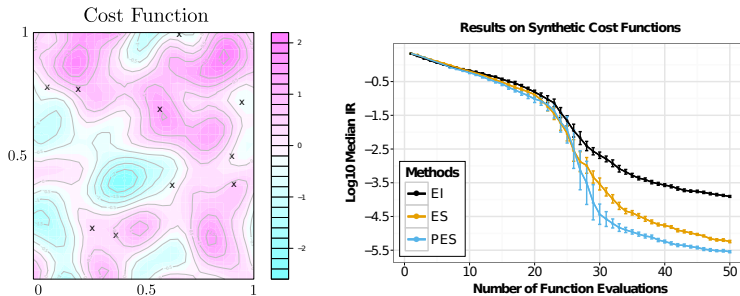

Figure 2: Left, example of objective functions $f$. Right, median of the immediate regret (IR) for the methods PES, ES and EI in the experiments with synthetic objective functions.

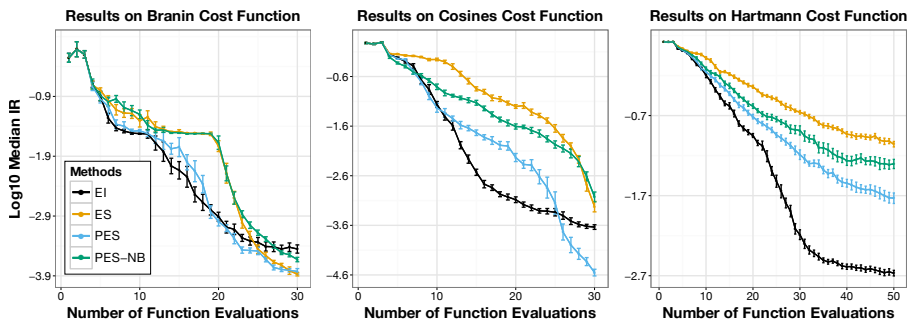

Figure 3: Median of the immediate regret (IR) for the methods EI, ES, PES and PES-NB in the experiments with well-known synthetic benchmark functions.

because the empirical distribution of IR values is very heavy-tailed. In this case, the median is more representative of the exact location of the bulk of the data. These results indicate that the best method in this setting is PES, which significantly outperforms ES and EI. The plot also shows that in this case ES is significantly better than EI.

We perform another series of experiments in which we optimize well-known synthetic benchmark functions including a mixture of cosines [2] and Branin-Hoo (both functions defined in $[0,1]^2$) as well as the Hartmann-6 (defined in $[0,1]^6$) [15]. In all instances, we fix the measurement noise to $\sigma^2 = 10^{-3}$. For both PES and EI we marginalize the hyperparameters $\psi$ using the approach described in Section 2.3. ES, by contrast, cannot average its approximation of (2) over the posterior on $\psi$. Instead, ES works by fixing $\psi$ to an estimate of its posterior mean (obtained using slice sampling) [27]. To evaluate the gains produced by the fully Bayesian treatment of $\psi$ in PES, we also compare with a version of PES (PES-NB) which performs the same non-Bayesian (NB) treatment of $\psi$ as ES. In PES-NB we use a single fixed hyperparameter as in previous sections with value given by the posterior mean of $\psi$. All the methods are initialized with three random measurements collected using latin hypercube sampling [5].

The plots in Figure 3 show the median IR obtained by each method on each function across 250 random initializations. Overall, PES is better than PES-NB and ES. Furthermore, PES-NB is also significantly better than ES in most of the cases. These results show that the fully Bayesian treatment of $\psi$ in PES is advantageous and that PES can produce better approximations than ES. Note that PES performs better than EI in the Branin and cosines functions, while EI is significantly better on the Hartmann problem. This appears to be due to the fact that entropy-based strategies explore more aggressively which in higher-dimensional spaces takes more iterations. The Hartmann problem, however, is a relatively simple problem and as a result the comparatively more greedy behavior of EI does not result in significant adverse consequences. Note that the synthetic functions optimized in the previous experiment were much more multimodal that the ones considered here.

## 3.1 Experiments with real-world functions

We finally optimize different real-world cost functions. The first one (NNet) returns the predictive accuracy of a neural network on a random train/test partition of the Boston Housing dataset [3].

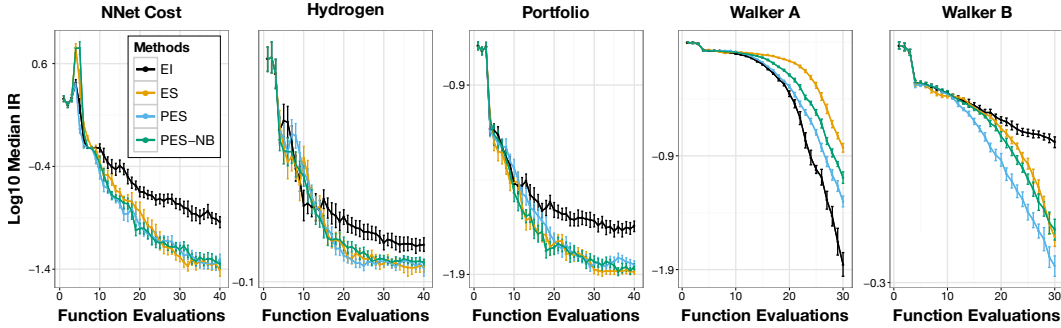

Figure 4: Median of the immediate regret (IR) for the methods PES, PES-NB, ES and EI in the experiments with non-analytic real-world cost functions.

The variables to optimize are the weight-decay parameter and the number of training iterations for the neural network. The second function (Hydrogen) returns the amount of hydrogen production of a particular bacteria in terms of the PH and Nitrogen levels of the growth medium [7]. The third one (Portfolio) returns the ratio of the mean and the standard deviation (the Sharpe ratio) of the 1-year ahead returns generated by simulations from a multivariate time-series model that is adjusted to the daily returns of stocks AXP, BA and HD. The time-series model is formed by univariate GARCH models connected with a Student's $t$ copula [12]. These three functions (NNet, Hydrogen and Portfolio) have as domain $[0, 1]^2$. Furthermore, in these examples, the ground truth function that we want to optimize is unknown and is only available through noisy measurements. To obtain a ground truth, we approximate each cost function as the predictive distribution of a GP that is adjusted to data sampled from the original function (1000 uniform samples for NNet and Portfolio and all the available data for Hydrogen [7]). Finally, we also consider another real-world function that returns the walking speed of a bipedal robot [30]. This function is defined in $[0, 1]^8$ and its inputs are the parameters of the robot's controller. In this case the ground truth function is noiseless and can be exactly evaluated through expensive numerical simulation. We consider two versions of this problem (Walker A) with zero-mean, additive noise of $\sigma = 0.01$ and (Walker B) with $\sigma = 0.1$.

Figure 4 shows the median IR values obtained by each method on each function across 250 random initializations, except in Hydrogen where we used 500 due to its higher level of noise. Overall, PES, ES and PES-NB perform similarly in NNet, Hydrogen and Portfolio. EI performs rather poorly in these first three functions. This method seems to make excessively greedy decisions and fails to explore the search space enough. This strategy seems to be advantageous in Walker A, where EI obtains the best results. By contrast, PES, ES and PES-NB tend to explore more in this latter dataset. This leads to worse results than those of EI. Nevertheless, PES is significantly better than PES-NB and ES in both Walker datasets and better than EI in the noisier Walker B. In this case, the fully Bayesian treatment of hyper-parameters performed by PES produces improvements in performance.

## 4 Conclusions

We have proposed a novel information-theoretic approach for Bayesian optimization. Our method, predictive entropy search (PES), greedily maximizes the amount of one-step information on the location $\mathbf{x}_\star$ of the global maximum using its posterior differential entropy. Since this objective function is intractable, PES approximates the original objective using a reparameterization that measures entropy in the posterior predictive distribution of the function evaluations. PES produces more accurate approximations than Entropy Search (ES), a method based on the original, non-transformed acquisition function. Furthermore, PES can easily marginalize its approximation with respect to the posterior distribution of its hyper-parameters, while ES cannot. Experiments with synthetic and real-world functions show that PES often outperforms ES in terms of immediate regret. In these experiments, we also observe that PES often produces better results than expected improvement (EI), a popular heuristic for Bayesian optimization. EI often seems to make excessively greedy decisions, while PES tends to explore more. As a result, EI seems to perform better for simple objective functions while often getting stuck with noisier objectives or for functions with many modes.

**Acknowledgements** J.M.H.L acknowledges support from the Rafael del Pino Foundation.

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
