[Supplementary Material]

# Supplementary material for: Predictive Entropy Search for Efficient Global Optimization of Black-box Functions

José Miguel Hernández-Lobato, Matthew W. Hoffman and Zoubin Ghahramani

## A   Details on approximating GP sample paths

In this section we give further details about the approach used in Section 2.1 to approximate a GP using random features. These random features can be used to approximate sample paths from the GP posterior. By optimizing these sample paths we obtain posterior samples over the global maxima $\mathbf{x}_\star$. We derive in more detail the kernel approximation from (5). Formally, the theorem of [4] states

**Theorem 1** (Bochner's theorem). *A continuous, shift-invariant kernel is positive definite if and only if it is the Fourier transform of a non-negative, finite measure.*

As a result given some kernel $k(\mathbf{x}, \mathbf{x}') = k(\mathbf{x} - \mathbf{x}', \mathbf{0})$ there must exist an associated density $s(\mathbf{w})$, known as its *spectral density*, which is the Fourier dual of $k$. This can be written as

$$k(\mathbf{x}, \mathbf{x}') = \int e^{-i\mathbf{w}^\mathsf{T}(\mathbf{x}-\mathbf{x}')} s(\mathbf{w}) \, d\mathbf{w},$$

$$s(\mathbf{w}) = \frac{1}{(2\pi)^d} \int e^{i\mathbf{w}^\mathsf{T}\boldsymbol{\tau}} k(\boldsymbol{\tau}, \mathbf{0}) \, d\boldsymbol{\tau}.$$

Further, we can treat this measure as a probability density $p(\mathbf{w}) = s(\mathbf{w})/\alpha$ where $\alpha = \int s(\mathbf{w}) \, d\mathbf{w}$ is the normalizing constant. Consequently, the kernel can be written as

$$k(\mathbf{x}, \mathbf{x}') = \alpha \, \mathbb{E}_{p(\mathbf{w})}[e^{-i\mathbf{w}^\mathsf{T}(\mathbf{x}-\mathbf{x}')}]$$

and due to the symmetry of $p(\mathbf{w})$ [see 22] we can write the expectation as

$$= \alpha \, \mathbb{E}_{p(\mathbf{w})}[\tfrac{1}{2}(e^{-i\mathbf{w}^\mathsf{T}(\mathbf{x}-\mathbf{x}')} + e^{i\mathbf{w}^\mathsf{T}(\mathbf{x}-\mathbf{x}')})]$$
$$= \alpha \, \mathbb{E}_{p(\mathbf{w})}[\cos(\mathbf{w}^\mathsf{T}\mathbf{x} - \mathbf{w}^\mathsf{T}\mathbf{x}')] \, .$$

We can then note that $\int_0^{2\pi} \cos(a + 2b) \, db = 0$ for any constant offset $a \in \mathbb{R}$. As a result, for $b$ uniformly distributed between $0$ and $2\pi$ we can write

$$= \alpha \, \mathbb{E}_{p(\mathbf{w})}[\cos(\mathbf{w}^\mathsf{T}\mathbf{x} - \mathbf{w}^\mathsf{T}\mathbf{x}') + \mathbb{E}_{p(b)}[\cos(\mathbf{w}^\mathsf{T}\mathbf{x} + \mathbf{w}^\mathsf{T}\mathbf{x}' + 2b)]]$$
$$= \alpha \, \mathbb{E}_{p(\mathbf{w},b)}[\cos(\mathbf{w}^\mathsf{T}\mathbf{x} + b - \mathbf{w}^\mathsf{T}\mathbf{x}' - b) + \cos(\mathbf{w}^\mathsf{T}\mathbf{x} + b + \mathbf{w}^\mathsf{T}\mathbf{x}' + b)]$$
$$= 2\alpha \, \mathbb{E}_{p(\mathbf{w},b)}[\cos(\mathbf{w}^\mathsf{T}\mathbf{x} + b) \cos(\mathbf{w}^\mathsf{T}\mathbf{x}' + b)] \, .$$

The last equality can be derived from the sum of angles formula, which leads to the identity: $2\cos(x)\cos(y) = \cos(x - y) + \cos(x + y)$. Finally, we can average over $m$ weights and phases

$$= \frac{2\alpha}{m} \, \mathbb{E}_{p(\mathbf{W},\mathbf{b})}[\cos(\mathbf{W}\mathbf{x} + \mathbf{b})^\mathsf{T} \cos(\mathbf{W}\mathbf{x}' + \mathbf{b})] \, ,$$

where $[\mathbf{W}]_i \sim p(\mathbf{w})$ and $[\mathbf{b}]_i \sim p(b)$ are stacked versions of the original random variables. The resulting quantity has the same expectation but results in a lower variance estimator. If we let $\phi(\mathbf{x}) = \sqrt{2\alpha/m} \cos(\mathbf{W}\mathbf{x} + \mathbf{b})$ denote a random $m$-dimensional feature generated by this model we can also write the kernel as $k(\mathbf{x}, \mathbf{x}') = \mathbb{E}_{p(\phi)}[\phi(\mathbf{x})^\mathsf{T}\phi(\mathbf{x}')]$.

We now briefly show the equivalence between a Bayesian linear model using random features $\phi$ and a GP with kernel $k$. Consider now a linear model $f(\mathbf{x}) = \phi(\mathbf{x})^\mathsf{T}\boldsymbol{\theta}$ where $\boldsymbol{\theta} \sim \mathcal{N}(\mathbf{0}, \mathbf{I})$ has a standard Gaussian distribution and observations $\mathcal{D}_n = \{(\mathbf{x}_i, y_i)\}_{i \leq n}$ of the form $y_i \sim \mathcal{N}(f(\mathbf{x}_i), \sigma^2)$. The posterior of $\boldsymbol{\theta}$ given $(\mathcal{D}_n, \phi)$ is also normal $\mathcal{N}(\mathbf{m}, \mathbf{V})$ where

$$\mathbf{m} = (\mathbf{\Phi}^\mathsf{T}\mathbf{\Phi} + \sigma^2\mathbf{I})^{-1}\mathbf{\Phi}^\mathsf{T}\mathbf{y},$$
$$\mathbf{V} = (\mathbf{\Phi}^\mathsf{T}\mathbf{\Phi} + \sigma^2\mathbf{I})^{-1}\sigma^2,$$

and where $[\mathbf{\Phi}]_i = \phi(\mathbf{x}_i)$ and $[\mathbf{y}]_i = y_i$ consist of the stacked features and observations respectively. We can also easily write the predictive distribution over $f$ evaluated at a test point $\mathbf{x}$, which is Gaussian distributed with mean and variance given by

$$\mu_n(\mathbf{x}) = \phi(\mathbf{x})^{\mathsf{T}}(\mathbf{\Phi}^{\mathsf{T}}\mathbf{\Phi} + \sigma^2\mathbf{I})^{-1}\mathbf{\Phi}^{\mathsf{T}}\mathbf{y},$$
$$v_n(\mathbf{x}) = \phi(\mathbf{x})^{\mathsf{T}}(\mathbf{\Phi}^{\mathsf{T}}\mathbf{\Phi} + \sigma^2\mathbf{I})^{-1}\phi(\mathbf{x})\sigma^2.$$

By a simple application of the matrix-inversion lemma these quantities can be rewritten in terms which only make use of the inner products between features,

$$\mu_n(\mathbf{x}) = \phi(\mathbf{x})^{\mathsf{T}}\mathbf{\Phi}^{\mathsf{T}}(\mathbf{\Phi}\mathbf{\Phi}^{\mathsf{T}} + \sigma^2\mathbf{I})^{-1}\mathbf{y}_{1:t},$$
$$v_n(\mathbf{x}) = \phi(\mathbf{x})^{\mathsf{T}}\phi(\mathbf{x}) - \phi(\mathbf{x})^{\mathsf{T}}\mathbf{\Phi}^{\mathsf{T}}(\mathbf{\Phi}\mathbf{\Phi}^{\mathsf{T}} + \sigma^2\mathbf{I})^{-1}\mathbf{\Phi}\phi(\mathbf{x}),$$

the expectations of which are equivalent to the kernel $k$ and we obtain the same expressions as that in (1).

# B   Details on approximating the predictive variance

We now provide further details on approximating the predictive variance $v_n(\mathbf{x}|\mathbf{x}_\star)$ of inputs $\mathbf{x}$ given the position of the global optimizer $\mathbf{x}_\star$. In particular we include all steps omitted in the presentation of Section 2.2.

## B.1   Incorporating the analytic latent constraints (C1.1)

We first turn to the random variables

$$\mathbf{z} = [f(\mathbf{x}_\star); \, \mathrm{diag}[\nabla^2 f(\mathbf{x}_\star)]],$$
$$\mathbf{c} = [\mathbf{y}_n; \nabla f(\mathbf{x}_\star); \mathrm{upper}[\nabla^2 f(\mathbf{x}_\star)]] = [\mathbf{y}_n; \mathbf{0}; \mathbf{0}].$$

Here $\mathbf{c}$ contains the random variables that we will condition on in order to enforce constraint C1.1. Given the input locations $\mathbf{x}$ and $\mathbf{x}_\star$ we can construct a kernel matrix $\mathbf{K}$ containing the covariance evaluated on the stacked vector $[\mathbf{z}; \mathbf{c}]$. We again refer to [25] in constructing this matrix which includes derivative observations, the computations of which are tedious but not overly complicated. Note also that the portions of $\mathbf{K}$ which correspond to $y_i$ will have an additional $\sigma^2$ due to the observation noise. Next let $\mathbf{K_z}$, $\mathbf{K_c}$, and $\mathbf{K_{zc}}$ denote the corresponding diagonal and off-diagonal blocks of the kernel matrix. We can now condition on the observed values of $\mathbf{c}$ to write

$$p(\mathbf{z}|\mathcal{D}_n, \text{C1.1}) = p(\mathbf{z}|\mathbf{c}) = \mathcal{N}(\mathbf{z}|\mathbf{m}_0, \mathbf{V}_0)$$

where $\mathbf{m}_0 = \mathbf{K_{zc}}\mathbf{K_c}^{-1}\mathbf{c}$ and $\mathbf{V}_0 = \mathbf{K_z} - \mathbf{K_{zc}}\mathbf{K_c}^{-1}\mathbf{K_{zc}^{\mathsf{T}}}$.

## B.2   Incorporating the non-analytic latent constraints (C1.2 and C2)

The additional constraints C1.2 and C2 can be introduced explicitly as in (6), which takes the form of a single Gaussian factor and $d+1$ non-Gaussian factors

$$p(\mathbf{z}|\mathcal{D}_n, \text{C1}, \text{C2}) \propto \mathcal{N}(\mathbf{z}|\mathbf{m}_0, \mathbf{V}_0)\Big[\prod_{i=1}^{d+1} t_i(z_i)\Big].$$

We approximate this distribution using a single multivariate Gaussian $q(\mathbf{z})$ where each non-Gaussian factor is replaced by a Gaussian approximation $\widetilde{t}_i(z_i) = \mathcal{N}(z_i; \widetilde{m}_i, \widetilde{v}_i)$ such that

$$q(\mathbf{z}) = \mathcal{N}(\mathbf{z}|\mathbf{m}, \mathbf{V}) \propto \mathcal{N}(\mathbf{z}|\mathbf{m}_0, \mathbf{V}_0)\Big[\prod_{i=1}^{d+1} \mathcal{N}(z_i; \widetilde{m}_i, \widetilde{v}_i)\Big]$$

where this approximation is parameterized by $\mathbf{m} = \mathbf{V}[\widetilde{\mathbf{V}}^{-1}\widetilde{\mathbf{m}} + \mathbf{V}_0^{-1}\mathbf{m}_0]$ and $\mathbf{V} = (\widetilde{\mathbf{V}}^{-1} + \mathbf{V}_0^{-1})^{-1}$. The parameters of the approximate factors are combined to form the vector $[\widetilde{\mathbf{m}}]_i = \widetilde{m}_i$ and the diagonal matrix $[\widetilde{\mathbf{V}}]_{ii} = \widetilde{v}_i$.

To compute the approximate factors we use expectation propagation (EP). EP is a procedure that starts from some initial values for the approximate factors $(\widetilde{m}_i, \widetilde{v}_i)$ and iteratively refines these quantities; here we initialize $\widetilde{m}_i = 0$ and $\widetilde{v}_i = \infty$ which corresponds to $\mathbf{m} = \mathbf{m}_0$ and $\mathbf{V} = \mathbf{V}_0$. At each iteration, for every factor $i$, we remove the contribution of the $i$th approximate factor to form the *cavity* distribution $q_{\setminus i}(\mathbf{z}) \propto q(\mathbf{z})/\widetilde{t}_i(z_i)$. Given the independent factors we consider here we can focus on each individual component $q_{\setminus i}(z_i)$ separately with mean and variance

$$\bar{m}_i = \bar{v}_i(m_i/v_{ii} - \widetilde{m}_i/\widetilde{v}_i),$$
$$\bar{v}_i = (v_{ii}^{-1} - \widetilde{v}_i^{-1})^{-1}.$$

Let $\hat{q}(z_i) \propto q_{\setminus i}(z_i)t_i(z_i)$ denote the *tilted* distribution where the $i$th approximate factor has been replaced by the corresponding real factor. EP proceeds by finding the approximation $q_i$ that minimizes the KL-divergence $D[\hat{q}_i || q_i]$ where $q_i$ is restricted to be Gaussian. This amounts to matching the first two moments. Finally, by removing the influence of the cavity distribution and setting $\widetilde{t}_i(z_i) \propto q_i(z_i)/q_{\setminus i}(z_i)$ we can update the approximate factors. This can be performed using the same procedure which forms the cavity distribution.

For both sets of constraints used in this work the moments can easily be obtained by computing the log normalizing constant $\bar{Z}_i = \log \int \mathcal{N}(z_i|\bar{m}_i, \bar{v}_i) \, t_i(z_i) \, dz_i$ and using the following identities:

$$\mathbb{E}_{\hat{q}}[z_i] = \bar{m}_i + \bar{v}_i \frac{\partial \bar{Z}_i}{\partial \bar{m}_i}, \qquad\qquad \text{Var}_{\hat{q}}[z_i] = \bar{v}_i - \bar{v}_i^2 \left( \left[\frac{\partial \bar{Z}_i}{\partial \bar{m}_i}\right]^2 - 2\frac{\partial \bar{Z}_i}{\partial \bar{v}_i} \right). \qquad (11)$$

We first show the updates for the parameters $(\widetilde{m}_i, \widetilde{v}_i)$ of the factors corresponding to constraints on the diagonal Hessian, i.e. where $t_i(z_i) = \mathbb{I}[z_i < 0]$. Given the moments of the tilted distribution in (11), we can remove the contribution of the cavity distribution as above and write

$$\widetilde{m}_i \leftarrow \bar{m}_i + \kappa^{-1}, \qquad\qquad \text{where } \alpha = -\frac{\bar{m}_i}{\sqrt{\bar{v}_i}},$$

$$\widetilde{v}_i \leftarrow \beta^{-1} - \bar{v}_i, \qquad\qquad \beta = \frac{\phi(\alpha)}{\Phi(\alpha)} \left[ \frac{\phi(\alpha)}{\Phi(\alpha)} + \alpha \right] \frac{1}{\bar{v}_i},$$

$$\kappa = -\left[ \frac{\phi(\alpha)}{\Phi(\alpha)} + \alpha \right] \frac{1}{\sqrt{\bar{v}_i}},$$

where $\phi$ and $\Phi$ are the standard Gaussian density and cumulative distribution functions, respectively. For the final soft-maximum constraint, $\Phi\big((z_i - y_{\max})/\sigma\big)$, the moments can be calculated in a similar fashion. Using the same procedure as above we arrive at very similar updates:

$$\widetilde{m}_i \leftarrow \bar{m}_i + \kappa^{-1}, \qquad\qquad \text{where } \alpha = \frac{\bar{m}_i - y_{\max}}{\sqrt{\bar{v}_i + \sigma^2}},$$

$$\widetilde{v}_i \leftarrow \beta^{-1} - \bar{v}_i, \qquad\qquad \beta = \frac{\phi(\alpha)}{\Phi(\alpha)} \left[ \frac{\phi(\alpha)}{\Phi(\alpha)} + \alpha \right] \frac{1}{\bar{v}_i + \sigma^2},$$

$$\kappa = \left[ \frac{\phi(\alpha)}{\Phi(\alpha)} + \alpha \right] \frac{1}{\sqrt{\bar{v}_i + \sigma^2}}.$$

### B.3 Incorporating the prediction constraint (C3)

Given some test input $\mathbf{x}$ we now turn to the problem of making predictions about $f(\mathbf{x})$. We again note that both the "prior" terms $\mathbf{m}_0$, $\mathbf{V}_0$ and the EP factors, $\widetilde{\mathbf{m}}$ and $\widetilde{\mathbf{V}}$, are independent of $\mathbf{x}$ and can be precomputed once for later use at prediction time.

Let $\mathbf{f} = [f(\mathbf{x}); f(\mathbf{x}_\star)]$ be a vector given by the concatenation of the latent function at $\mathbf{x}$ and $\mathbf{x}_\star$. The distribution for $\mathbf{f}$ given the first two constraints can be written as

$$p(\mathbf{f}|\mathcal{D}_n, \mathrm{C1}, \mathrm{C2}) \approx \int p(\mathbf{f}|\mathbf{z}, \mathbf{c}) \, q(\mathbf{z}) \, d\mathbf{z} = \mathcal{N}(\mathbf{f}|\mathbf{m_f}, \mathbf{V_f}). \qquad (12)$$

By writing $p(\mathbf{f}|\mathbf{z}, \mathbf{c})$ above we are assuming that $\mathbf{f}$ is independent of C1.2 and C2 given $\mathbf{z}$ and as a result the above is simply an integral over the product of two Gaussians. Let $\mathbf{K}_\dagger$ be the cross-covariance matrix evaluated between $\mathbf{f}$ and $[\mathbf{z}; \mathbf{c}]$ and $\mathbf{K_f}$ the covariance matrix associated with $\mathbf{f}$.

The posterior above will then be Gaussian with mean and variance

$$\mathbf{m_f} = \mathbf{K}_\dagger [\mathbf{K} + \widetilde{\mathbf{W}}]^{-1} [\mathbf{c}; \widetilde{\mathbf{m}}]$$
$$\mathbf{V_f} = \mathbf{K_f} - \mathbf{K}_\dagger [\mathbf{K} + \widetilde{\mathbf{W}}]^{-1} \mathbf{K}_\dagger^\mathsf{T},$$

where $\widetilde{\mathbf{W}}$ is a block-diagonal matrix where the first block is zero and the second is $\widetilde{\mathbf{V}}$ (note this matrix is also diagonal since $\widetilde{\mathbf{V}}$ is diagonal).

The constraint that $f(\mathbf{x}) < f(\mathbf{x}_\star)$ can be incorporated by introducing an additional factor ensuring that $\mathbf{c}^\mathsf{T}\mathbf{f}$ is positive for $\mathbf{c} = [-1; 1]$. The resulting density has a log normalizing constant given by

$$Z = \log \int \mathcal{N}(\mathbf{f}|\mathbf{m_f}, \mathbf{V_f}) \, \mathbb{I}[\mathbf{c}^\mathsf{T}\mathbf{f} > 0] \, d\mathbf{f}$$

$$= \log \int \mathcal{N}(a|m, v) \, \mathbb{I}[a > 0] \, da = \log \Phi(m/\sqrt{v}).$$

where $m = \mathbf{c}^\mathsf{T}\mathbf{m_f}$ and $v = \mathbf{c}^\mathsf{T}\mathbf{V_f}\mathbf{c}$. The derivatives of $Z$ with respect to $\mathbf{m_f}$ and $\mathbf{V_f}$ can then be taken and a multivariate form of (11) can be used to obtain the first two moments [18]. In particular, the second moment is given by

$$\mathrm{Var}[\mathbf{f}] = \mathbf{V_f} - v^{-1}\beta(\alpha + \beta)(\mathbf{V_f}\mathbf{c}\mathbf{c}^\mathsf{T}\mathbf{V_f}),$$

where $\alpha = m/\sqrt{v}$ and $\beta = \phi(\alpha)/\Phi(\alpha)$. Finally, the first diagonal component of the resulting matrix corresponds to the marginal variance of $f(\mathbf{x})$, which can be simplified as in (9). This marginal variance can be used as the variance parameter in a Gaussian approximation to $p(f(\mathbf{x})|\mathcal{D}_n, \mathbf{x}_\star)$. For the PES acquisition function we only need this variance parameter since the entropy of a Gaussian does not depend on its first moment.