[Reviews · NeurIPS 2014]

Submitted by Assigned_Reviewer_10

The paper under review 'Predictive Entropy Search for Efficient Global Optimization of Black-box Functions' presents an approach for Bayesian optimization that measures entropy in the posterior predictive distribution. The authors present a method named Predictive Entropy Search (PES) that is derived by a reparameterization of the expected information gain based on the symmetry of mutual information. The authors describe the posterior sampling, predictive entropy approximation and hyper-parameter learning steps of PES. The advantages of the proposed method are that it is more efficient and accurate than alternatives and that it allows to handle the hyper-parameters in a fully Bayesian way. Finally, a comparison of the performance of the proposed method with 'Entropy Search' and 'Expected Improvement' on different real world cost functions is presented. The results show that PES performs best on some of the tasks. The quality of the presented paper is very high. The paper is clearly written and has a sound mathematical formulation.

Some minor concerns:
* The difference to the related method 'Entropy Search' should be made a bit clearer and presented in more detail.
* In the evaluation the computational time of the different methods should be provided.
Summary: The paper "Predictive Entropy Search for Efficient Global Optimization of Black-box Functions" under review presents a new acquisition function for Bayesian optimization using the posterior differential entropy in a mathematical novel and computational efficient manner. The results demonstrate that this approach has the potential of replacing current state of the art acquisition functions in Bayesian optimization.

Submitted by Assigned_Reviewer_24

[Review update: based on the author's response and other reviewers comments I feel my initial review was too harsh - I have updated my score accordingly now.]

This paper proposes a variation on Bayesian/Gaussian Process driven black-box optimization. The main novelty is the use of a entropy-based criterion for selecting points to evaluation during optimization. This is developed into a practical implementation and experimental results are presented. The work is interesting, relevant to NIPS and for the most part clearly presented.

References are adequate, though not exhaustive. The work in this area has sometimed contained large overlaps. I would not agree with some of the attribution to the literature (e.g. Mockus et al I don't think can claim to propose "expected improvement").

To implement the approach, a sequence of approximations is required (see 2.1 - 2.3). It is not clear how well each of these affects the results. The evaluation in Fig.1 goes some way however towards validating the implementation. The experimental results are mixed. The problems tested are of relatively low dimensionality. It is not clear whether the technique will scale well to larger problems. It would be very interesting for example to see the authors apply the algorithm to the BBOB benchmarking set: http://coco.gforge.inria.fr/doku.php Nevertheless some of the experimental results do show good comparative performance for the proposed technique.

Other comments:

- Typo: paper line 221: "right-hand size" -> "right-hand side"
- paper line 086, it is not clear to me why you say that the above described aquisition functions are "based on optimistic estimates".
Summary: A new Bayesian algorithm for black-box global optimization, which uses a entropy-based function to control the selection of future search points. The function makes sense but the approach has many caveats and approximiations required to implement it.

Submitted by Assigned_Reviewer_32

This manuscript proposes a new acquisition function for use in Bayesian optimization with a Gaussian process belief on a latent objective function. The proposed criterion is a rewriting of the previously suggested entropy search (ES) objective [Hennig and Schuler, JMLR 2012], which seeks to maximize the posterior entropy of the _location_ of the global maximum x*, H[x* | D]. We sequentially greedily select the next point x' in the domain of the objective which maximizes the expected posterior entropy H[x* | D'], where D' is the current data augmented by the proposed new observation (x', y'). This requires marginalizing the unknown observation value y' and, if we are to be fully Bayesian, the hyperparameters of the model, \theta. The authors first write out the mutual information between x* and y' (given D) in the way that it would most often be encountered in an active learning context (and how entropy search was originally posed by Hennig and Schuler). The expression is unwieldy due to the marginalization of y' and the repeated need to approximate H[x* | D'], which requires many approximations.

The authors instead propose to use a trick that has been previously used in other similar contexts, e.g., [Houlsby, et al. NIPS 2012]: they rewrite the mutual information so that only _marginal_ entropies of y' are required. These are often much easier to work with. The new objective is H[y' | x', D] - E[ H[y' | x', D, x*] ], where the expectation is taken over x*. The former is simply the predictive entropy of a GP, which can be computed in closed form assuming fixed hyperparameters. The second term is more novel and interesting; we must approximate the predictive uncertainty of y', conditioning the GP on the location of the global maximizer x*. Conditioning a GP on x* is intriguing, and the bulk of the paper is about an approximate method for doing so. First, the authors propose a method for sampling from the current distribution p(x* | D). The proposed method for doing so exploits a now-common trick of approximating the covariance from samples in the Fourier domain. This gives each sampled function a finite representation, allowing us to find its maximum easily (although the authors omit details on this point). Given a location x*, they must condition the GP on knowledge of its location. This would be rather difficult (it essentially requires a censored observation at every point in the domain), so the authors instead propose a set of relaxed but reasonable requirements:

* x* must at least be a local maximum. This is easy to accomplish in the GP framework by conditioning (analytically) on observations forcing the gradient to be zero and the Hessian to be negative definite. The latter could be difficult in high dimension, so we instead condition H[f] at x* to be diagonal with negative entries.
* The value at x* must exceed all those previously seen.
* The observed value y' must be less extreme than the value y* we selected subject to the previous constraint. This reduces the global maximizer condition to just apply at the point x' under consideration, effectively truncating the predictive distribution p(y' | x', D, x*) at y*.

The first condition can be analytically handled in the GP framework, but the truncation required by the second is not tractable, so the authors rely on the well-known expectation propagation algorithm for approximating the joint distribution between y* and the diagonal of the Hessian at x*. Finally, the authors handle uncertainty in the hyperparameters by marginalizing the acquisition function over \theta; this is done by simply slice sampling from p(\theta | D). A series of experiments indicate that the proposed method (called "predictive" entropy search, but it's really just a new approximation to the same entropy search method) works well. In particular, Figure 1 is quite convincing at showing that the PES approximation is much closer to the true objective (approximated by rejection sampling) than the ES approximation. The experiments are well designed and explained, and code will be released upon publication. Although PES appears to be usually better than ES, it still loses to simpler methods like expected improvement on some problems.

One philosophical issue I have with entropy search is that it's not entirely clear to me that H[x* | D] is somehow a "better" objective than, say, EI. The scenario I always picture is one where we have determined that the optimum of a function must lie in a low, quite flat, but quite broad valley. In this case we can be very unsure about the exact location of x*, but can be quite certain about its corresponding value y*. Do I really always need to locate x* with perfect accuracy if I can guarantee that I can find a point with x with f(x) within \epsilon of y* = f(x*)? Note that a score such as EI can indicate clearly to me that I expect with high certainty not to find a better point and therefore can stop now. Further, a lookahead version of EI is in fact the Bayesian optimal policy to the optimization problem with simple regret; see [Gruenewaelder, et al. AISTATS 2010]. [Osborne, et al. LION 2009] used a two-step lookahead version of EI that, although prohibitively expensive, encouraged more exploration than the standard, myopic EI criterion and worked better in practice. I know that this criticism of entropy search is not unique to me, but I have yet to see someone take it head-on and offer a frank discussion on the point.

Overall, I like the paper and think the rewriting of the ES objective, together with the interesting and reasonable approximations made by the authors to derive the PES variant, deserve to be published and discussed further. I also look forward to an eventual code release.

Some minor notes:

- There are many stacked parens in this paper that are exactly the same height; for an example, see the extreme right-hand-side of (2). It would help readability (at least for me) if some judicious use of \big, \Big, etc. were used.
- Page 1 line 35 (1/35): You say that the derivatives of f are unavailable, but this does not strictly need to be the case. Indeed you rely on the ease with which we can condition a GP on derivative observations later. I could imagine a scenario where it might be useful to condition a function on a derivative observation, perhaps by a finite-different approximation as in [Osborne, et al. LION 2009].
- 1/52: A comma after "decision" would make this more clear.
- 2/80: maxima -> maximum
- 6/306: I don't know where the location (0.572, 0.687) is!
- 8/426: "hyperparameters" has not been hyphenated until now.
Summary: This paper provides an interesting rewriting of entropy search that appears much more useful in practice than the previous approximation. It would definitely be of interest to the NIPS community.
Author Feedback
Author rebuttal: We thank the reviewers for their interesting comments and careful analysis. We are also very pleased to receive some strong positive feedback. Below you can find our response to some of the issues raised by the reviewers:

Reviewer_24:

"Incremental work and unlikely to have much impact":

As the other two reviewers point out, we believe that our work is different from typical submissions and can have a major impact in the field of Bayesian optimization. Our work presents a novel acquisition function based on the expected reduction of posterior entropy. Entropy-based approaches are among the most successful methods for Bayesian active learning. However, their applicability to the field of Bayesian optimization has been limited so far because of the difficulty of computing the corresponding acquisition function. The main novelty of our work lays in a new mathematical and computationally efficient manner for doing this. We believe our approach can have a big impact by facilitating the development of new state-of-the-art Bayesian optimization methods based on the reduction of posterior entropy.

"The approach has many caveats and approximations required to implement it":

Our method uses several approximations, but these are based on well understood methods that are routinely used for solving machine learning problems. In particular,

1 - We approximate the kernel function using a discrete representation in terms of basis functions. See, "Random features for large-scale kernel machines" in NIPS, 2007. (280 citations in Google scholar). This technique is a common approach used to scale up kernel methods.

2 - We use expectation propagation (EP) for approximate Bayesian inference. See, "Expectation propagation for approximate Bayesian inference", in UAI 2001, (705 citations in Google scholar). EP is routinely used in industry and academia and it is known to provide very accurate approximations in related problems such as binary classification with Gaussian processes.

As indicated by reviewer_32, Figure 1 is quite convincing at showing the quality of our approximations. This figure shows that our approximation to the true acquisition function is much better than other alternatives such as the approximation used by the method Entropy Search.

"The problems tested are of relatively low dimensionality, not clear whether the technique will scale well to larger problems":

We include in our experiments results on optimization problems of moderate dimensionality, 6 and 8 dimensions in the problems Hartman and Walker A and B. Our experiments show that entropy based approaches can produce state-of-the-art solutions in these settings when the amount of measurement noise is high. Note that derivative-free, global optimization of general, noisy cost functions in more than 10-20 dimensions is well-known to be difficult problems within the time horizons that we consider here.

“I would not agree with some of the attribution to the literature (Mockus et al do not propose EI)”:

We include a reference to Mockus as an early originator of the terminology for “Bayesian optimization” and developer of a method very similar to EI. We would credit the more modern usage of EI with the EGO work of Jones et al. We will clarify this.

"BBOB benchmarking set":

We thank the reviewer for pointing us to this benchmarking set which was unknown to us. We will include experiments using this set in an extended version of the paper.

"Acquisition functions based on optimistic estimates":

Here we were referring mostly to approaches such as UCB which are based on the principle of “optimism in the face of uncertainty”. Approaches such as EI and PI are based more on probabilistic measures of improvement, which while related, are not quite the same. We will clarify this.

Reviewer_32:

"Philosophical issue, better objective than EI":

We agree with the reviewer on this. We do not claim that our method is always better than EI. The optimal acquisition function is going to depend always on the characteristics of the function being optimized and on the loss function being used to evaluate performance. The reviewer also makes very good “philosophical” points that we hope to expand upon in the final draft. Briefly:

1. As the reviewer also points out it definitely is the case that both PES and EI are myopic approximations to the full-lookahead procedure.

2. Further, we could also easily modify our formulation to only seek an “epsilon-optimal” point.

3. An advantage of PES (and other information-based approaches) over EI is that EI will only evaluate points that provide improvement, whereas PES can evaluate points that provide information about the location of the maximum without necessarily improving over the current best estimate. This may prove quite useful in more structured probabilistic models, although we leave that for future work.

"derivative of f unavailable, but this does not need to be the case":

Our method could be easily adapted to use derivative observations, as indicated by the reviewer.

“I do not know where (0.572, 0.687) is”:

We will mark this location in the figure.

Reviewer_10:

"Make clear difference with respect to Entropy Search":

The objective for both PES and ES is the same. We only differ in our approximations and in that our formulation of the objective allows for integrating over the hyperparameters. We will clarify this in the final draft.

"Computational time of the different methods":

All methods are of the same order of complexity and are primarily constrained by the computation required for performing inference with the GP and integrating out the hyperparameters. We will also clarify this.